# Experiencing nature leads to healthier food choices
Maria Langlois [1] ✉ & Pierre Chandon [2]

Experiencing nature has been linked to a host of benefits for health and well-being. Here, we examine if exposure to nature influences the food choices that may contribute to nature's benefits. Five between-subject experiments ($n = 39$, $n = 698$, $n = 885$, $n = 1191$, and $n = 913$) show that individuals exposed to the natural environment choose healthier foods when compared to those exposed to urban environments or a control condition. Nature's effects are observed for various foods and beverages, across samples from three countries, and in varied contexts, such as taking a walk in a park (vs. a city street) and looking at photos of nature (vs. urban or control) scenes. These findings provide insights into the relationship between proximity to nature and health.

Investigating the health effects of experiencing nature—and the consequences associated with the lack thereof—has acquired significant importance given that children and adults now spend a majority of their time indoors and far less time outdoors than previous generations[1–3], where 68% of the world's population is expected to live in urban areas by 2050, up from 30% in 1950[4]. The health benefits of exposure to nature are only now beginning to make their way into the scientific literature, and an account for the sources of these effects remains indeterminate. This research provides insights into the matter by demonstrating that exposure to nature (vs. urban and control) environments promotes the choice of healthier (more nutritious and less processed) foods, an established foundation for health.

A natural environment can be defined as one without human intrusion or intervention[5,6], measured on a spectrum with a primeval forest on one end and a wholly man-made urban setting on the other. Between these two extremes, natural elements may be incorporated within an urban environment. For instance, tree-lined streets and city parks are sufficient forms of nature to reap the benefits of experiencing nature[7,8]. Spending time in nature is associated with psychological, cognitive, physiological, social, spiritual, and medical benefits, such as lower rates of obesity[9,10]. Recent correlational studies have also demonstrated that higher levels of connectedness to nature are associated with increased fruit and vegetable intake as well as with greater dietary diversity[11]. While these studies advocate for exposure to nature as a healthy eating intervention, their results may be driven by self-selection.

A few experimental studies explored the impact of exposure to nature on nonfood-related delay-discounting tasks, such as financial trade-offs[12,13]. To date, few studies (listed in Supplementary Table 1) have explored the impact of exposure to nature on dietary choice decisions through experimental methods. Most of these studies focused on the effects of exposure to nature on impulsive decision-making, typically measured via delay-discounting tasks and food desirability scales[14,15]. The few studies that examined eating directly focused on highly specific decisions, such as the amount of sugar added to bubble tea by participants who had the intention to lose weight[14], rather than the more generic food choices made by "normal" (e.g., non-dieting) eaters for typical meals. Other studies[16] examined self-reported dietary recall, such as the number of vegetable servings consumed over time, but did not examine the effects of exposure to nature per se but the effects of a family-based training program that increases feelings of connectedness to nature. A few studies[17,18] examined the particular case of stress eating, yielding contradictory findings and leaving the question of the effects of exposure to nature on typical eating unresolved.

The strongest evidence for the effects of nature exposure comes from two field studies. The first study[19] found that placing posters depicting a nature scene next to a vending machine increased the sales of healthier snacks when compared to posters depicting a fair with carousels, or no poster at all. However, this study did not examine the effects of posters illustrating urban scenes, and it examined aggregate sales as opposed to individual choices. Consequently, it cannot be determined whether the poster with the nature scene attracted more health-conscious buyers or if it truly increased preferences for healthier snacks at the individual level. The second study[20] created a nature ambiance in a restaurant through changes in lighting, images, and sounds, but found only a marginally significant effect on the choice of vegetarian options compared to the old ambiance and no credible evidence of a difference when compared to a fast-food ambiance.

Overall, existing research is very limited and has yet to conclusively determine if exposure to nature can lead to healthier eating in the general

[1]Assistant Professor of Marketing at the Cox School of Business, Southern Methodist University, Dallas, TX, USA. [2]L'Oréal Chaired Professor of Marketing, Innovation and Creativity and the director of the INSEAD Sorbonne University Behavioral Lab at INSEAD, Boulevard de Constance, 77300 Fontainebleau, France. ✉e-mail: Langlois@smu.edu

population and when choosing from a variety of foods and beverages (as a snack or as a full meal). To achieve this goal, it is imperative to conduct studies that involve consequential food choices and actual eating behaviors, rather than solely relying on self-reported food desirability or food craving scales, as is predominantly used in the literature thus far. Research is also needed to understand if exposure to nature encourages the choice of truly healthy, nutritious, and unprocessed foods like fruits and vegetables or if it merely promotes the consumption of processed foods positioned as healthy through marketing claims like "diet" or "light". Beyond food choice, the impact of exposure to nature on overall food consumption quantity requires investigating as well, as it remains unclear whether experiencing nature leads people to eat healthier in terms of food quality and/or if it impacts food quantity. Finally, it is important to know whether it is exposure to a natural environment that leads to healthier food choices or whether it is exposure to an urban environment that encourages unhealthier choices.

Through a series of five between-subjects experiments (see Table 1), we test the hypothesis that experiencing nature in the outdoors (Study 1) or through a virtual nature scene (Studies 2–5) leads people to make healthier food choices when compared to experiencing an urban environment (Studies 1–5) or a control environment (Study 2). Because exposure to nature increases feelings of connectedness to nature[16], which is associated with healthy attitudes like increased respect for one's body[21] and healthier dietary choices[11], we hypothesize that nature exposure specifically increases the motivation to eat more healthily, and not simply to lose weight for appearances' sake. Consequently, experiencing nature should lead to healthier food choices by increasing the importance placed on perceived food healthiness in these choices—and not by altering perceptions of food healthiness (Study 3)—should not influence total food quantity intake (Study 1), nor preferences for foods marketed as diet nor light (Study 4). Finally, Study 5 explores prior inconclusive results[15] that may have been driven by a lack of power and/or the reliance on indirect measures of food desirability rather than more direct measures of food choices.

## Methods

All the online studies were pre-registered, and their sample size was determined based on power analyses[22]. Each study featured a distinct sample of participants—and participant demographic variables, such as age and sex, were deemed tangential to the phenomenon, and therefore were not

assessed. We obtained internal review board approval from the university ethics committee at INSEAD, received informed consent from all participants, and complied with all relevant ethical regulations.

### Study 1

Study 1 comprised of a field experiment (without pre-registration), which took place during the spring of 2016 at a university and cultural center located directly across from a large public park in Paris, France. Forty-three Parisian residents were recruited for this study, and the study was conducted with one participant at a time. We instructed participants to take a 20-min walk on a pre-specified route, either in a large local park (nature condition) or on nearby city streets (urban condition). Both routes were of similar length, distance, and difficulty, and had the same starting point. Participants were asked to refrain from eating two hours prior to the experiment, as well as from listening to music or engaging in other activities during their walk. One participant was not able to complete the assignment because it started raining and three served as initial trial participants. This left $N = 39$ participants (25 female) for the analyses (20 in the nature condition and 19 in the urban condition), and thus 312 observations, since there were 8 food options per participant. Participants completed their walk independently with the guidance of a map (available on ResearchBox). Both the nature (i.e., local park) and urban (i.e., city street) maps illustrated three landmarks along the routes: at the beginning, middle, and end, where participants were instructed to take photos to be shown to the researcher upon their return. The photography task, inspired by prior research[23], served as a cover story and as a manipulation check, indicating that each participant completed the entire route as instructed.

After the walk, participants had access to a snack buffet for 10 min, which was framed as compensation for their participation in the study. The buffet consisted of four healthy snacks (bananas, apples, dried fruits, and mixed nuts) and four unhealthy snacks (strawberry cookies, apricot cookies, potato chips, and brownies) pre-tested for healthiness, as detailed in Supplementary Note 2. Based on the pre-test evaluations (and homogeneity across nutritional profiles), the strawberry and apricot cookies (same brand and type of cookie) were aggregated during the coding phase, creating 7 overall food choice options in the data, from the 8 original choice options. All of the snacks were laid out visibly on a large table in the experiment room, with numerous quantities of each snack available. Pre-packaged

## Table 1 | Summary of studies

| Study | Nature intervention | Outcome variable(s) | Summarized findings |
|---|---|---|---|
| 1 (N = 39) | Outdoor field study with 20-minute nature vs. urban walks | Food choice and consumption in a snack buffet that comprised of healthy snacks and unhealthy snacks, pre-tested for healthiness | Participants asked to take a nature walk ate healthier snacks at the buffet when compared to those who went on the urban walk. There was no evidence of a difference in the quantity of snacks consumed between the two conditions. |
| 2 (N = 698) | Pre-registered online study with photos of a hotel room window view of nature vs. urban vs. control (i.e., closed curtain) scenes | Choice of lunch from a room service menu with 12 food/beverage options (4 mains, 4 sides, 4 drinks), pre-tested for healthiness | When compared to participants in both the urban and control conditions, participants in the nature group made significantly healthier lunch choices. Meanwhile, there was no significant difference between participants in the urban and control conditions in their choice of healthy foods. |
| 3 (N = 885) | Pre-registered online study with the same nature and urban photos as Study 2 (but without the window frames) | Choice of lunch from a room service menu with 12 food/beverage options (4 mains, 4 sides, 4 drinks), with respondents' own food/beverage healthiness ratings | Participants in the nature condition were significantly more likely to choose healthier foods when compared to participants in the urban condition. Respondents' own food healthiness ratings were more predictive of food choice in the nature condition than in the urban condition. |
| 4 (N = 1191) | Pre-registered online study with photos of nature vs. urban scenes taken by the same photographer | Incentive-compatible snack choice task between: (1) a natural, healthy snack, (2) a tasty, indulgent snack, or (3) a diet, light snack | Participants in the nature condition were more likely to select a natural, healthy snack and less likely to select a tasty, indulgent snack or a diet, light snack than participants in the urban condition. |
| 5 (N = 913) | Pre-registered online study with photos of nature and urban scenes used in a past inconclusive study[15] | Choice of lunch from a room service menu with 12 food/beverage options (4 mains, 4 sides, 4 drinks), pre-tested for healthiness | Regardless of how food healthiness was measured, participants in the nature condition made significantly healthier food choices when compared to participants in the urban condition. |

foods were either already portioned and packaged out of the box, or portioned out by the researcher and packaged in small, sealed clear bags based on the portion sizes indicated on the package. In accordance with FDA standards[24], each individual fruit was considered one serving. Participants were informed that they could have any and as many snacks as they wanted, but that all snacks had to be consumed on site and could not be saved for later. Participants were told to leave their trash in the room, which was cleaned once the experiment ended. The researcher timed each buffet session and returned to the experiment room once the 10 min had passed so that participants could consume their desired snacks without external influence. After participants completed the experiment and were debriefed and escorted from the lab, the researcher took inventory and recorded the snack choices and quantities consumed for each participant.

## Study 2

705 American residents were recruited online through Prolific Academic. The number of respondents was determined to achieve 95% power with a two-sided α = 0.05 based on the results of Study 1—and the study was pre-registered (https://aspredicted.org/YDK_47B). After pre-registered exclusions, such as attention check failures, this left a sample size of 698 participants (231 in the control condition, 233 in the nature condition, and 234 in the urban condition). In this study, participants were randomly assigned between subjects to one of three conditions: nature, control, or urban. Participants read about a scenario in which they had recently won a radio sweepstakes contest for a free night at a hotel. They were instructed to imagine being in their hotel room with an illustrated window view from their room, which depicted either a nature scene, an urban scene, or the same window with the curtains closed (in the control condition). Respondents were asked to pay careful attention to the photo and to write a sentence describing the scene depicted in it. Afterwards, participants were asked to select their choice of lunch, comprising of one main dish, one side dish, and one beverage, that they would order through room service, a food choice task was adapted from prior research[25]. The room service menu consisted of 12 food/beverage items (4 mains, 4 sides, 4 drinks), which were pre-tested (see Supplementary Note 3) for healthiness using continuous and categorical measures. For this study, the two unhealthy main courses were a peanut butter and jelly sandwich and a hot dog, while the two healthy main courses comprised of a salmon salad and a cobb salad. The four sides consisted of a Kit Kat bar (unhealthy), potato chips (unhealthy), a fruit salad (healthy), and unsweetened Greek yogurt (healthy). The four beverages were regular Coke (unhealthy), Mountain Dew (unhealthy), coconut water (healthy), and mineral water (healthy). Participants had a total of 12 different food/beverage options and made one selection for each of the three categories: one main dish, one side dish, and one beverage, in order to construct their desired meal.

## Study 3

920 American residents were recruited online via Prolific Academic. The number of respondents was determined to achieve 95% power with a two-sided α = 0.05 based on the results of Study 2, as detailed in the pre-registration (available at https://aspredicted.org/CR8_K2H). After pre-registered exclusions, the remaining sample comprised of 885 participants (442 in the nature condition and 443 in the urban condition). In this two by two between-subjects design study, participants were randomly assigned to one of the two environmental conditions (nature or urban) as well as to one of the measurement orders (food choice and then healthiness ratings, or healthiness ratings and then food choice). This second factor allows us to measure potential carryover effects of measurement, whereby some participants would choose healthier foods simply because they had been asked to rate the foods' healthiness prior to making their selection (vs. after they have made their choice). Our hypothesis is that the healthiness ratings will be a stronger predictor of choice in the nature condition than in the urban condition, independent of the measurement order.

As in Study 2, participants were told to imagine that they had recently won a day trip to a special location depicted in the accompanying photo.

Based on the condition, they either saw the nature photo or the urban photo used in Study 2, but without the window frames. Participants in both groups were instructed to reflect on their day trip in the location illustrated in the photo and to pay careful attention to the scene. To ensure that participants were paying attention to the photos presented, they were asked to describe the scene depicted in one sentence. Participants in the "choice and then healthiness rating" condition were then asked to select food and beverage items that would serve as a packed lunch if they were to go on a day trip to the location depicted. In making their food and beverage selections for the packed lunch, participants chose one of the four possible beverage, main course, and side dish options used in Study 2. Finally, the participants were presented with those same four beverage, main course, and side dish options once again, and were asked to rate each item on a 7-point Likert scale from "extremely unhealthy" to "extremely healthy". The order of the food choice and healthiness rating tasks was reversed in the "healthiness rating and then choice" order condition.

## Study 4

In this study, 1200 American residents were recruited online through Prolific Academic. The number of respondents recruited was determined to achieve 95% power with a two-sided α = 0.05 based on the results of Study 2. As pre-registered (https://aspredicted.org/W97_91]), 9 participants were excluded for ineligible device use and attention check failures, leaving 1191 participants in the sample (598 in the nature condition and 593 in the urban condition). Study 4 used a between-subjects design with random assignment to either the nature or urban condition. Participants were instructed to carefully observe the scene presented, as they would be asked memory-based questions about the scene later in the survey. To control for the quality and style of the photos, the nature and urban photos were taken by the same photographer[26]. Then, to ensure that respondents paid attention to the scene, they were asked to view the photo for at least 15 s and to describe the scene in one or two sentences. Finally, to increase the generalizability of the task, Study 4 did not include photos of specific foods but instead asked participants to choose between three textual descriptions of snacks: "a natural, healthy snack", "a tasty, indulgent snack", or "a diet, light snack". These textual descriptions were tested to ensure that participants perceive the snack described as the "natural, healthy snack" as healthiest (Supplementary Note 4). For incentive compatibility, participants were informed that 10 individuals would be randomly selected to receive the snack they had chosen free of charge. For logistical reasons, these 10 participants were compensated financially for their selection.

## Study 5

As with the preceding studies, the number of participants was determined to achieve 95% power with a two-sided α = 0.05, based on the results of Study 2. 920 participants were recruited for pre-registered Study 5 (https://aspredicted.org/DBR_HP4). 913 participants (455 in the nature condition and 458 in the urban condition) remained in the analysis after pre-registered exclusions for ineligible device use and attention check failures. To test the robustness of the effects across populations, the participants in this study were based in the United Kingdom—whereas our past samples came from the United States (Studies 2–4) and France (Study 1). As in studies 2 and 3, participants were told to imagine that they had recently won a free night at a nice hotel, where an accompanying photo depicted the window view from their hotel room. The photographic stimuli for this study were taken from another paper[15], where participants in the nature condition viewed a photo of a waterfront view with green cliffs, while participants in the urban condition saw a photo of a modern building in a clean city without people. The photo task was framed as a memory and visual perception task in preparation for memory-based questions later in the survey. Next, participants were asked to describe the photographic scene in one sentence. As with Studies 2 and 3, participants were instructed to select food and beverage items for lunch, with a choice of one of four possible beverages, main courses, and side dish options.

## Results

### Study 1

Parisian residents were randomly assigned (in individual sessions) to either take a 20-min walk in a park with abundant nature or to a comparable walk on city streets. After the walk, participants were invited to eat from a snack buffet containing 4 healthy foods (different types of fruits and nuts) and 4 unhealthy foods (chips, brownies, and cookies—the strawberry and apricot cookies were collapsed for a resulting total of 3 unhealthy foods instead of 4). The type and amount of food consumed were recorded. As shown in Fig. 1, the nature intervention did not influence the total quantity of food consumed ($M_{nature} = 2.58$ servings, $SD_{nature} = 1.41$ vs. $M_{urban} = 2.70$ servings, $SD_{urban} = 1.08$; $F(1, 37) = 0.10$, $p = 0.76$, $\eta^2 = 0.003$). However, it did influence the type of food consumed. Compared to those that walked in the city, the participants who walked in the park consumed more servings of healthy foods ($M_{nature} = 1.80$ healthy servings, $SD_{nature} = 1.28$ vs. $M_{urban} = 1.05$ healthy servings, $SD_{urban} = 0.85$; $F(1, 37) = 4.56$, $p = 0.04$, $\eta^2 = 0.110$) and fewer servings of unhealthy foods

($M_{nature} = 0.78$ unhealthy servings, $SD_{nature} = 0.84$ vs. $M_{urban} = 1.65$ unhealthy servings, $SD_{urban} = 1.38$; $F(1, 37) = 5.82$, $p = 0.02$, $\eta^2 = 0.136$).

To further test the effects of the intervention on the consumption of healthy and unhealthy foods, we estimated a random effect model in SPSS[27] using the MIXED procedure, which allows for correlated errors at the participant level to account for the fact that each participant provided 8 observations (one per food; collapsed to 7 to account for homogeneity across the strawberry and apricot cookies). Thus, we regressed the number of servings consumed on two binary variables, NATURE (coded as ½ for the nature walk and -½ for the urban walk), HEALTHINESS (coded as ½ for the four healthy foods and -½ otherwise) and their interaction. Data distribution was assumed to be normal, but this was not formally tested. Neither the main effect of NATURE nor of HEALTHINESS were statistically significant (respectively: $B = -0.05$, $t = -0.89$, $p = 0.38$ and $B = -0.05$, $t = -0.58$, $p = 0.56$). However, their interaction was statistically significant ($B = 0.48$, $t = 2.93$, $p < 0.01$), indicating that, compared to the urban walk, participants in the nature walk condition consumed more healthy food and less unhealthy food.

**Fig. 1 | Food choices after exposure to natural, urban, or control scenes. a** Number of servings of healthy snacks, unhealthy snacks, and total number of servings consumed by participants assigned to a nature walk or to an urban walk in Study 1.
**b** Percentage of healthy (vs. unhealthy) food choices made by participants exposed to an image of a natural, control, or urban setting in Study 2.
**c** Percentage of healthy (vs. unhealthy) food choices made by participants exposed to an image of a natural, control, or urban setting in Study 3.
**d** Unstandardized regression coefficients of subjective food healthiness scores in a conditional logistic regression of food choice in Study 3.
**e** Percentage of participants that selected a 'tasty, indulgent snack' vs. 'a diet, light snack' vs. 'a natural, healthy snack', when exposed to an image of a natural or urban setting in Study 4. **f** Percentage of healthy (vs. unhealthy) food choices made by participants exposed to an image of a natural or urban setting in Study 5. All error bars denote standard errors.

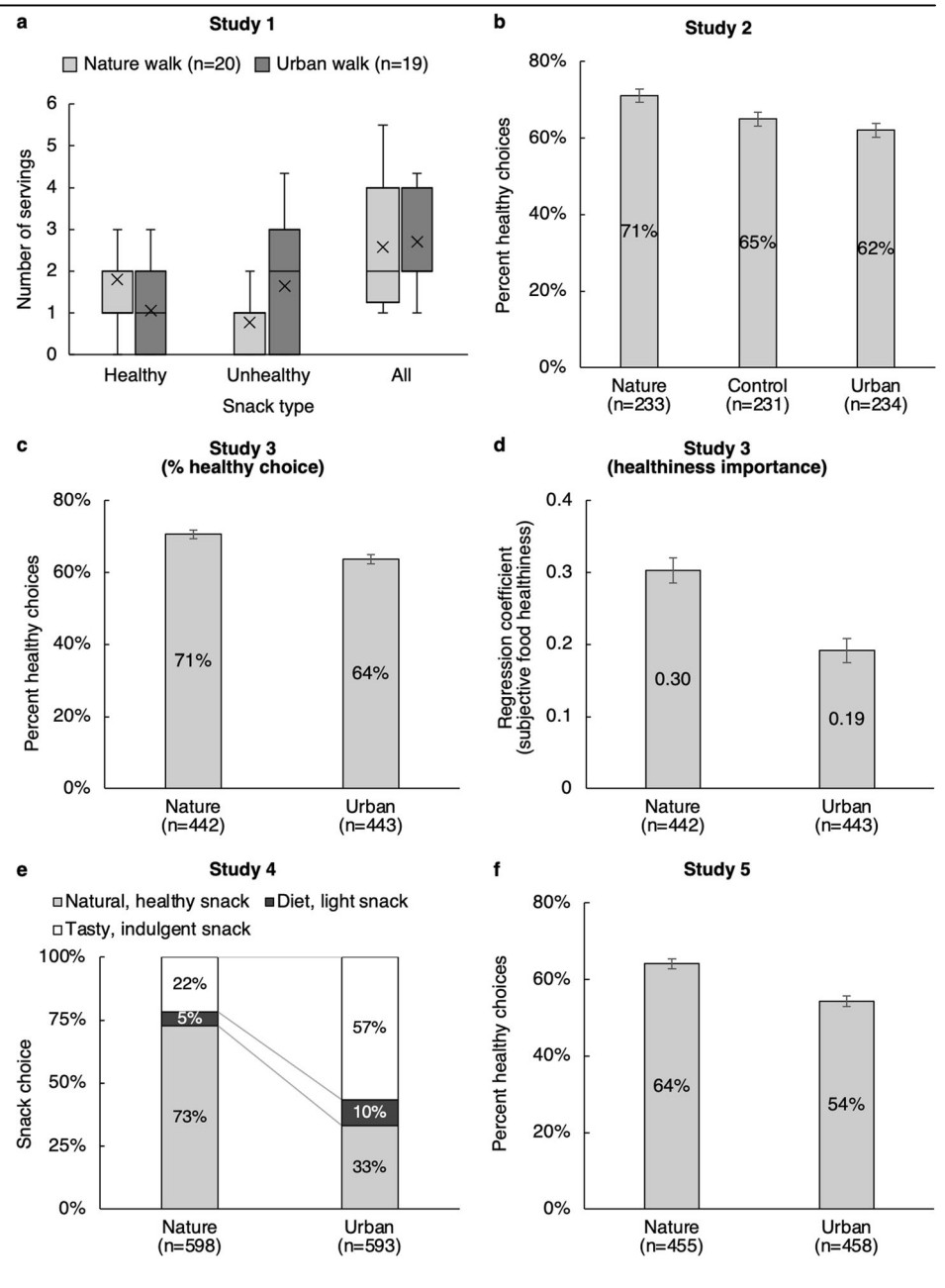

To provide a comparison to the other studies that did not measure consumption quantity, we also analyzed the effects of the intervention on the selection of food consumed, regardless of the quantity consumed. In the urban condition, 47.4% of the chosen foods were healthy and 52.6% were unhealthy. In contrast, in the nature condition, 71.7% of the selected foods were healthy and 28.3% were unhealthy, a statistically significant improvement, $\chi^2(1, 84) = 5.18, p = 0.02$. Study 1 provides field evidence that a nature walk leads people to choose healthy snacks over unhealthy ones. However, it cannot determine whether it was exposure to the natural environment that led participants to consume healthy food, or whether it was exposure to the urban environment that led to the consumption of unhealthy food. Study 2 addresses this question by incorporating a neutral control condition. Additionally, subsequent studies examine whether the effects of nature exposure can be obtained by exposure to photos of natural or urban scenes, without necessitating an actual walk in a natural or urban environment.

## Study 2

American respondents were instructed to imagine being in their hotel room with a window view onto either a nature scene, an urban scene, or the same window with the curtains closed. Afterward, participants were asked to select the room service meal that they would eat from a lunch menu that featured photos of four main courses, four side dishes, and four beverages. Half of the options were pre-tested to be healthy and the other half unhealthy. As pre-registered, we analyzed the three choices comprising the meal—the main course, side dish, and beverage—together, after accounting for the repeated nature of the data. In the urban condition, 62% of the selected foods were healthy, and in the control condition, 65% of the selected foods were healthy; that proportion increased to 71% in the nature condition, $\chi^2(2, 2094) = 13.76, p < 0.01$. As pre-registered, we first conducted binary logistic regressions with the selection of each of the 12 foods or beverages (coded as 1 if chosen and 0 otherwise) as the dependent variable, yielding 8376 observations (12 observations per participant). As in Study 1, HEALTHINESS was coded as ½ for healthy foods and −½ for unhealthy foods. Using the control condition of the intervention as the default level, we created two binary variables to capture the effects of the intervention: NATUREVCONTROL (coded as ⅔ in the nature condition and −⅓ otherwise) and URBANVCONTROL (coded as ⅔ in the urban condition and −⅓ otherwise). We also added the interaction of food healthiness with these two binary variables.

There was a statistically significant main effect of HEALTHINESS ($B = 0.89$, $Wald = 280.06, p < 0.01$), indicating that participants were more likely to choose the healthier options over the unhealthy ones, overall. The main effects of NATUREVCONTROL and URBANVCONTROL were not statistically significant (respectively, $B = −0.05$, $Wald = 0.46$, $p = 0.50$ and $B = 0.02$, $Wald = 0.08, p = 0.78$). More importantly, there was a significant interaction between NATUREVCONTROL and HEALTHINESS ($B = 0.35$, $Wald = 7.16, p < 0.01$), indicating that participants were more likely to select a healthy option in the nature (vs. control) condition, as predicted. In alignment with our hypothesis, the interaction between URBANVCONTROL and HEALTHINESS ($B = −0.19$, $Wald = 2.28, p = 0.13$) was not statistically significant. Finally, to compare the nature and urban conditions to one another, we estimated an additional binary logistic regression excluding the control condition, and with a new variable, NATUREVURBAN (coded as ½ in the nature condition and −½ in the urban condition). The interaction between NATUREVURBAN and HEALTHINESS was statistically significant ($B = 0.54$, $Wald = 17.53, p < 0.001$), indicating that, as in Study 1, participants were more likely to choose a healthy option in the nature condition than in the urban condition.

To account for the fact that each participant made three choices among four options per category—one for a main course, one for a side dish, and one for a beverage—rather than 12 independent food choices, we also estimated a conditional logistic regression to compare the effects of the food attributes (healthy vs. unhealthy) on the likelihood of choice across the three experimental conditions (nature, urban, & control). Using the CLOGIT procedure in STATA[28] with clustering at the participant level, we found a main effect of HEALTHINESS ($B = 0.67$, $z = 12.56, p < 0.001$), a statistically significant interaction between HEALTHINESS and NATUREVCONTROL ($B = 0.27$, $z = 2.03, p = 0.04$), and an insignificant interaction between HEALTHINESS and URBANVCONTROL ($B = −0.15, z = −1.15, p = 0.25$). Note that the main effect of the intervention is omitted in the conditional logistic regression because all participants must choose one of the four options regardless of whether they are in the nature, control, or urban condition. These interaction results were replicated with the continuous measure of food healthiness ratings provided by similar participants described in Supplementary Note 3 (see ResearchBox for details). Overall, the results are robust to the estimation method used.

By integrating a neutral control condition, Study 2 demonstrates that exposure to a natural environment drives healthy food choice, as opposed to the potential alternative explanation of exposure to an urban environment driving unhealthy food choice. Subsequent studies examine the hypothesized mechanism of action, which is that nature exposure increases the importance attached to health when making food choices.

## Study 3

Study 3 used the same procedure as Study 2 and the same nature and urban stimuli but measured each participant's evaluation of the healthiness of the food options, whereas preceding studies relied on an a priori categorization of food or used continuous healthiness ratings obtained from an external sample. The additional measure allows for examining whether nature exposure increases the importance of each participant's own perception of food healthiness in driving food choices. It also permits examining the alternative explanation that nature exposure influences perceived food healthiness, which could influence choices even if the importance of health remained constant. To remove concerns that the measurement of food healthiness might bias food choices, or vice versa, half of the participants rated foods on healthiness prior to making their meal selections, while the other half rated foods on healthiness after making their selections.

On average, participants in the nature condition made healthier food choices when compared to participants in the urban condition ($M_{nature} = 70.6\%$ healthy vs. $M_{urban} = 63.7\%$ healthy; $\chi^2(1, 2655) = 14.15$, $p < 0.001$). This result was obtained regardless of whether food choices were made before the healthiness ratings ($M_{nature} = 65.9\%$ healthy vs. $M_{urban} = 57.5\%$ healthy; $\chi^2(1, 1317) = 9.78$, $p < 0.01$) or after ($M_{nature} = 75.2\%$ healthy vs. $M_{urban} = 69.8\%$ healthy, $\chi^2(1, 1338) = 4.95$, $p = 0.03$). There was no statistically significant difference between the food ratings in the nature and urban conditions ($M_{nature} = 3.94$, $SD_{nature} = 2.25$ vs. $M_{urban} = 3.98$, $SD_{urban} = 2.22$; $F = 1.11, p = 0.29$).

To estimate whether exposure to nature increased the importance of health considerations, we estimated the same conditional logistic regression as in Study 2. The independent variables were HEALTHRATING (the participant's own mean-centered healthiness ratings), NATURE (equal to ½ in the nature condition and −½ in the urban condition), RATINGSFIRST (equal to ½ in the "rating then choice" condition and −½ otherwise), and their interactions. The main effect of NATURE and RATINGSFIRST are omitted because, by construction, they are constant across all food options for a given participant. The results revealed a significant main effect of HEALTHRATING ($B = 0.25, z = 20.64, p < 0.001$), indicating that participants were more likely to select foods or beverages that they had rated higher in terms of healthiness. More importantly, there was a statistically significant interaction of HEALTHRATING and NATURE ($B = 0.11, z = 4.58, p < 0.001$). This shows that, as expected, healthiness ratings were more predictive of choice in the nature condition than in the urban condition. The regression coefficient of the healthiness ratings was larger in the nature condition ($B = 0.30, SE = 0.02$) than in the urban condition ($B = 0.19, SE = 0.02$). Although tangential to our hypotheses, there was also a statistically significant interaction of HEALTHRATING and RATINGSFIRST ($B = 0.11, z = 4.53, p < 0.001$), indicating that healthiness ratings were more predictive of choice when they were collected before the food choices. More importantly, the three-way interaction between NATURE, RATINGSFIRST, and HEALTHRATING was not statistically significant ($B = −0.03, z = −0.52, p = 0.60$).

Study 3 demonstrates that experiencing nature drives individuals to make healthier food choices—and participants' healthier choices become more aligned with their own perceptions of food healthiness. By capturing each participant's own healthiness ratings, Study 3 rules out the notion that experiencing nature leads people to choose foods that are simply perceived as healthier by most people but not necessarily by themselves. Rather, healthfulness is sought out based on one's own perception of what is healthy. It also rules out the alternative explanation that nature exposure merely makes healthy foods appear healthier since healthiness ratings were unchanged regardless of whether people were in the nature or urban condition.

Incorporating individuals' own healthiness ratings raises the question of the heterogeneity in people's own definitions of food healthiness. Multiple and non-mutually exclusive interpretations of food healthiness coexist. For example, low-calorie "diet" food and unprocessed nutritious foods are similarly marketed as "healthy", even though prior research has documented notable differences in food choices depending on whether a person's motivation is health per se or the desire to lose weight[29,30]. Study 4, therefore, distinguishes between the goal to eat healthfully by choosing unprocessed, nutritious food and the goal of managing one's weight by eating low-calorie "diet" food, while continuing to offer the option to focus on taste, which remains the number one driver of eating motivation[31,32]. To test the robustness of the effects of experiencing nature, Study 4 implements an incentive-compatible procedure with consequential decisions—where participants are incentivized to select foods based on their own true preferences, as there is a chance they will win the food they select—and measures eating goals rather than the choice of specific foods.

## Study 4

Consistent with the preceding online studies, American participants were asked to view a photographic scene of a nature view or an urban view, which were taken by the same professional photographer[26] to minimize quality and style differences. After the intervention, participants indicated their choice of either (1) a natural, healthy snack, (2) a diet, light snack, or (3) a tasty, indulgent snack. To increase the consequentiality of their decisions, participants were informed that they could win their snack of choice.

Experiencing nature significantly increased the importance that respondents placed on food healthiness compared to the other eating goals. In the nature condition, 72.9% of the participants selected the natural, healthy snack, while only 33.2% of participants in the urban condition did so ($\chi^2(1, 1191) = 188.35$, $p < 0.001$). Importantly, exposure to nature reduced, rather than increased, preferences for a diet, light snack ($M_{nature} = 5.4\%$ vs. $M_{urban} = 10.3\%$, $\chi^2(1, 1191) = 10.08$, $p = 0.002$). Finally, exposure to nature also reduced preference for a tasty, indulgent snack ($M_{nature} = 21.7\%$ vs. $M_{urban} = 56.5\%$, $\chi^2(1, 1191) = 151.10$, $p < 0.001$).

In sum, Study 4 provides incentive-compatible evidence that exposure to nature gives rise specifically to healthy eating goals as opposed to motivating dieting behaviors. It also suggests that experiencing nature reduces the importance of taste goals. Given the robustness of these findings across the four preceding studies, one may wonder why some published studies failed to identify the same effect. We hypothesize that the null results in prior studies were driven by a lack of power or sensitivity in the measures that led to type II errors. To test this hypothesis, Study 5 used stimuli from a low-powered inconclusive study[15].

## Study 5

Study 5 was conducted using a nature scene photo and an urban scene photo from a study[15] that found inconclusive effects of nature exposure on fruit and vegetable desirability and null results on energy-dense food desirability. A power analysis led us to recruit 920 participants, 8.6 times as many as in the inconclusive study. Unlike the inconclusive study, which measured general food desirability on visual analog scales, Study 5 used the food choice task adapted from earlier research[25], used in Studies 2 and 3.

Consistent with our prior studies, there was a significant main effect of exposure to nature on food choices, where participants assigned to the nature condition were more likely to choose healthier options ($M_{nature} = 64.1\%$ vs. $M_{urban} = 54.3\%$; $\chi^2(1, 2739) = 27.27$, $p < 0.001$). The same conditional logistic regression used in Studies 2 and 3 showed a significant main effect of HEALTHINESS ($B = 0.38$, $z = 7.69$, $p < 0.001$), indicating that, on average, participants exhibited a greater preference for the healthier options. In addition, there was a significant positive interaction between HEALTHINESS and NATURE ($B = 0.41$, $z = 4.17$, $p < 0.01$). Similar conclusions were obtained when using the mean-centered continuous measure of food healthiness (HEALTHRATING) collected from another sample and described in Supplementary Note 3. There was a main effect of HEALTHRATING ($B = 0.12$, $z = 9.42$, $p < 0.01$) and a positive interaction between HEALTHRATING and NATURE ($B = 0.10$, $z = 3.90$, $p < 0.01$), indicating that exposure to nature increased the likelihood of choosing foods that are generally perceived as healthier. Overall, Study 5 demonstrates that the stimuli used in a previous study with inconclusive results reliably reproduce the effects of experiencing nature when the study is sufficiently powered.

## Discussion

The results of five experiments demonstrate that experiencing nature leads people to make healthier food choices than when experiencing a less natural, urban environment. This appears to be a robust effect; it was witnessed in food consumption decisions that took place at a snack buffet after outdoor walks and in online studies of incentive-compatible consumption intentions for entire meals following exposure to natural and urban scenes. These effects also hold across a variety of foods/beverages, contexts, and nationalities. Our replication efforts using published stimuli (in Study 5) demonstrate that previous inconclusive results were driven by a lack of statistical power or sensitivity in their measure of food preferences. Finally, these effects replicate in both a pre- and post/current-COVID world, as these studies spanned from 2016 (Study 1) to 2023 (Study 5). Despite external shocks and unprecedented circumstances, exposure to nature has proven effective in driving healthy food choices across populations and throughout the years.

Notably, we find that it is exposure to nature that drives healthier food choices rather than exposure to urban environments driving unhealthy food choices. Participants in the urban condition (with views of city streets) made unhealthy choices similar to those in the control condition (with a closed curtain window), where the environment was hidden from view. This could be because, at least in industrialized countries, most people live in man-made urban environments, which have become, de facto, the "normal" environment. Given prior results suggesting that feeling connected to nature matters more than just being exposed to it[33], it would be useful to determine the minimum levels of nature exposure necessary to benefit from it. Speaking to the mechanism of action, we found that experiencing nature increases the importance of health in driving food choices while decreasing preferences for reduced-calorie or indulgent foods. Furthermore, an implicit association task (Supplementary Note 5) demonstrates the implicit connection between nature and healthiness.

Nature exposure's influence on healthy eating is likely multiply-determined. Therefore, we recommend that future research test multiple potential mechanisms simultaneously to compare their importance and the conditions under which they operate. Future research is necessary to examine the role that affect, stress, priming, perceived restorativeness, delay discounting, and self-perception may play in explaining why nature exposure increases the motivation for healthy eating[34]. Additionally, we recommend that future work explore the boundary conditions associated with nature's effects on healthy eating. Supplementary Note 6 reports the results of a study, which finds a similar proportion of healthy food choice between exposure to nature versus urban scenes taken in the winter with snow, suggesting that greenery may play a role in nature's effects, although the characteristics of the landscape itself may also matter[17]. It would also be interesting to explore whether certain elements of vitality or awe could be altered to enhance or suppress nature's influence on food choice.

## Limitations

One of the weaknesses in our research is that we did not determine how long the benefits derived from experiencing nature endure. Another weakness is that we only studied the food choices made for a single consumption occasion, such as a snack or lunch. To address these concerns, it would be important to conduct longitudinal research into the effects of nature exposure on changes in diet over time. Future research should also examine whether these effects vary according to population characteristics. For example, one would expect nature exposure to be particularly beneficial to people living in urban environments compared to those living closer to nature in suburban or rural environments. We also recommend that future work challenge and explore what it means to eat "healthy" in this context, as elements like food quantity add an additional layer of nuance and understanding to healthy food choice. Developing inclusive interventions would be particularly important in this context, as research has identified disparities among several populations. For instance, research on green spaces in urban environments has revealed inequalities in access to nature for African American and Hispanic populations in the United States[35,36]. This is particularly alarming given that obesity in America has seen a disproportionate rise among African-American and Hispanic groups[37].

Not only do these findings have theoretical implications for the interdisciplinary study of food choices and the underlying decision-making processes, but they also provide practical insights for consumers, parents, food manufacturers, schools, and employers, who are invested in their own and possibly their children's, students', or employees' food choices. Stakeholders concerned with public health should pay particular attention to the health implications of urban planning and design. Additionally, companies could consider investing in green spaces, especially if they can be near workplace cafeterias. Given that social connections play an important role in the diffusion of healthy eating habits[38], bringing nature into school or workplace cafeterias where people eat in groups could be effective as well. Finally, marketers of healthier food products and alternatives could leverage images of nature in the advertising or on the packages of food products that are naturally healthier, such as fruits and vegetables. By demonstrating that experiencing nature promotes healthier food choices, our findings reveal a significant benefit provided to human societies by natural ecosystems—and help explain why proximity to nature is associated with good health and well-being.

## Data availability

Materials, pre-registrations, and data for all studies are available at https://researchbox.org/1674.

## Code availability

Code for statistical analyses for all studies can be found at https://researchbox.org/1674.

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

## Acknowledgements
The authors thank Alix Barasch, Amitava Chattopadhyay, Nofar Duani, Sung-Jin Jung, Nabila Langlois, Fabrice Le Lec, Hilke Plassmann, Manoj Thomas, and Wendy Wood for their comments.

## Author contributions
M.L. conceived of the presented idea and conducted the field experiment. M.L. and P.C. planned and carried out all other (online) experiments together. P.C. supervised M.L. throughout the project as part of M.L.'s doctoral dissertation. M.L. took the lead on writing the manuscript; both authors contributed to the manuscript.

## Competing interests
The authors declare no competing interests.
