## [Peer review file · Communications Psychology]

4th Sep 23

Dear Dr Langlois,

Thank you for your patience during the peer-review process. Your manuscript titled "Healthy by Nature: How Experiencing Nature Drives Healthy Food Choices" has now been seen by 3 reviewers, whose comments are appended below. You will see that they find your work of some potential interest. However, they have raised quite substantial concerns that must be addressed. In light of these comments, we cannot accept the manuscript for publication, but would be interested in considering a revised version that fully addresses these serious concerns.

We hope you will find the Reviewers' comments useful as you decide how to proceed. Should additional work allow you to address these criticisms, we would be happy to look at a substantially revised manuscript. If you choose to take up this option, please highlight all changes in the manuscript text file, and provide a detailed point-by-point reply to the reviewers.

Editorially, we consider that the revised manuscript should provide additional work to address reviewer #1's and reviewer #2's concerns about alternative explanations and boundary conditions. Also, we would suggest carefully revising the manuscript to remove any speculation about untested mechanisms or effects. In addition, for manuscripts that interpret null results, we require Bayes Factors or equivalence tests to interpret the null results. Please also ensure to use appropriate language to describe the results. Statements such as 'There is no difference between x and y.' or 'X does not affect Y.' must be revised to read 'We found [no/little] credible evidence of a difference between x and y.' or 'We found [no/little] credible evidence that X affects Y.'

If the revision process takes significantly longer than five months, we will be happy to reconsider your paper at a later date, provided it still presents a significant contribution to the literature at that stage.

Please use the following link to submit your revised manuscript, point-by-point response to the Reviewers' comments with a list of your changes to the manuscript text (which should be in a separate document to any cover letter) and any completed checklist:

[link redacted]

Please do not hesitate to contact me if you have any questions or would like to discuss the required revisions further. Thank you for the opportunity to review your work.

Best regards,

Hannah Hao

Hannah Hao, PhD
Editorial Board Member
Communications Psychology
orcid.org/0000-0002-3342-9132

EDITORIAL POLICIES AND FORMATTING

Editorial Policy: Policy requirements (Download the link to your computer as a PDF.)

Furthermore, please align your manuscript with our format requirements, which are summarized on the following checklist:

Communications Psychology formatting checklist

and also in our style and formatting guide Communications Psychology formatting guide .

* TRANSPARENT PEER REVIEW: Communications Psychology uses a transparent peer review system. This means that we publish the editorial decision letters including Reviewers' comments to the authors and the author rebuttal letters online as a supplementary peer review file. However, on author request, confidential information and data can be removed from the published reviewer reports and rebuttal letters prior to publication. If your manuscript has been previously reviewed at another journal, those Reviewers' comments would not form part of the published peer review file.

* CODE AVAILABILITY: All Communications Psychology manuscripts must include a section titled "Code Availability" at the end of the methods section. In the event of publication, we require that the custom analysis code supporting your conclusions is made available in a publicly accessible repository; please choose a repository that provides a DOI for the code; the link to the repository and the DOI must be included in the Code Availability statement. Publication as Supplementary Information will not suffice. We ask you to prepare and upload code at this stage, to avoid delays later on in the process.

* DATA AVAILABILITY:

All Communications Psychology research manuscripts must include a section titled "Data Availability" at the end of the Methods section or main text (if no Methods). More information on this policy, is available at <http://www.nature.com/authors/policies/data/data-availability-statements-data-citations.pdf>.

At a minimum the Data availability statement must explain how the data can be obtained and whether there are any restrictions on data sharing. Communications Psychology strongly endorses open sharing of data. If you do make your data openly available, please include in the statement:

We recommend submitting the data to discipline-specific, community-recognized repositories, where possible and a list of recommended repositories is provided at

<http://www.nature.com/sdata/policies/repositories>.

If a community resource is unavailable, data can be submitted to generalist repositories such as figshare or Dryad Digital Repository. Please provide a unique identifier for the data (for example a DOI or a permanent URL) in the data availability statement, if possible. If the repository does not provide identifiers, we encourage authors to supply the search terms that will return the data. For data that have been obtained from publicly available sources, please provide a URL and the specific data product name in the data availability statement. Data with a DOI should be further cited in the methods reference section.

REVIEWER EXPERTISE:

Reviewer #1 food decision-making

Reviewer #2 food decision-making

Reviewer #3 food decision making, nature/environment, self-regulation, and health

Reviewer #1 (Remarks to the Author):

This series of studies aims to examine whether being immersed in and exposed to images of nature, as opposed to urban environments, encourages adults to make healthier food choices. The researchers conclude that this is true across all of their studies. They also conclude that the mechanism for this is an increase in motivation to make healthy eating choices after exposure to nature (as opposed to reducing calories, and as opposed to urban settings increasing motivation to make unhealthy eating choices).

Results could have implications for our understanding of the health benefits of nature and possibly developing effective interventions for healthier eating down the line. The researchers include some careful controls in these studies as well. However, I would want to re-review the paper to determine its suitability for publication with a few questions answered first:

Intro:

I would take out the beginning quote and also the bit about fresh air in the Intro since that is not what is being measured here. Both walking conditions provided fresh air, and then viewing

conditions provided no fresh air, so there was no difference in fresh air exposure between environmental conditions. Plus, sometimes indoor air can also be healthier than outdoor air (ie wildfires of late), so this premise may be best to leave out altogether.

Instead of “regular” eating, maybe use “typical” eating? (Intro)

What is meant by “incentive-compatible food choice”?

Methods

Did the authors make any measurements of stimuli/environment restorativeness? Ie, did they disentangle environmental category (natural vs urban) from perceived restorativeness? (ie Berman et al., 2008; Kotabe et al., 2017, Mahamane et al., 2020). If not, there should be some discussion on how it may not be nature itself that motivates healthy eating decisions, but perhaps some other dimension(s) not yet measured or disentangled here (that may correlate with environmental category).

Similarly, since Nature, Urban, and Control were often represented by only one picture—how do the researchers know that it was the environmental *category* influencing decisions, and not the subject of the individual particular photo the researchers chose to represent the category (ie a lake representing Nature in Studies 2 and 3 vs the word “Stop” on concrete representing the category Urban?)

Also in Study 2/3, how long did the participants look at the photos for?

Rain was mentioned for one participant in Study 1—did participants generally experience the same (overall balance of) weather/temperature/time of day across conditions in Study 1? This would be important to know.

Results

In Study 3, how did perceived healthiness align with actual healthiness?

Discussion

Other possible mechanisms like priming, attention, emotion, prosociality, intention to not harm environment, self-controlled decision-making, etc need more discussion. There is a lot of research on how nature can affect these, and so it may not be easy to discount interconnected mechanisms.

I am not really sure what this series of studies can tell us about the importance of outdoor recess in schools. Some schools would have urban and some natural environments for recess, and effects of indoor environments were not tested here as a comparison.

In sum, this is an important, fascinating study on a topic that contains constructs often difficult to operationalize. The authors rightly point out limitations and shortcomings. Overcoming these limitations would certainly increase impact of the study, and yet, I understand this is outside of the scope of this particular and still impressive series of studies. Thus, in the end, I do believe this could be appropriate for publication with a sufficient future revision.

Reviewer #2 (Remarks to the Author):

In the manuscript “Healthy by Nature: How Experiencing Nature Drives Healthy Food Choices” 5 studies show that exposure to nature boost propensity to choose healthy foods. Even though other research has started to investigate the nature—healthy eating connection, the paper clearly describes how its methodological design elements (e.g., experimentation, urban and control comparison groups) add to existing studies on the general link between nature and healthy eating. On the one hand, the field study with real nature experiences and food choice is impressive, and yet on the other hand it’s also valuable to know that the effect generalizes even to “fake” nature exposure (e.g., pictures on a computer screen). This suggests solid potential practical usefulness. Aside from these positives, there are a couple of conceptual and empirical questions the manuscript would benefit from clarifying.

1. Analysis presentation and decisions could be clearer: It’s cumbersome to follow the analyses, perhaps because the designs are mixed and inherently more complex, with varying interdependencies (e.g., intervention is between-subjects but main/side/beverage choices are nested within-subject) but also because some statements describing what exact analyses were performed are vague (“accounting for the fact that each participant provided 8 observation” — how?). It may also be beneficial to explain the underlying rationales for choosing certain analysis strategies (e.g., what’s the advantage of treating each food choice as 0 vs. 1 in a design where people must choose exactly 1 from each category, compared to adding up the number of healthy items and running an ANOVA). To be clear, I’m NOT suggesting the authors chose inappropriate analysis strategies—I’m merely saying they could help readers better understand the value of the selected analyses.

2. Role of healthiness motives: Study 3 claims to show that “exposure to nature drives individuals to make healthier food choices by giving rise to a motive to eat more healthfully” (p. 10). Apologies if I’m overlooking something, motives (or goals or whatever construct best encapsulates people’s reasons for doing something) were not measured anywhere. If by “importance” the authors mean the higher regression coefficients, then the statement is a bit tautological, as betas definitionally encode importance (albeit statistical importance to the regression, not necessarily psychological importance to people’s motivations).

3. Distinction between healthy vs. low-calorie is unclear: Two aspects of the question if nature inspires a preference for healthier vs. lower-calorie (examined in Study 4) remained unclear: (a) What exactly do the authors mean by healthy when they juxtapose it with low-calorie? I understand and agree that healthy and low-calorie are not equal and definitely distinguishable (e.g., from a nutritional quality standpoint, a snack of raw almonds will almost certainly be deemed healthier despite being more calorific than a snack of sugar-free chocolate). But what do the authors mean: more nutrient-dense or (as some phrasing on p. 10 suggests) more natural, as opposed to less nutrient-dense or more processed? Or something else?

(b) Why do the authors feel this distinction is particularly important? Does it reveal something about the process? In the GD, one phrase seems to imply that this finding shows that the effect is “really” about healthy eating, not “just” about calorie reduction (p. 11, second to last paragraph). That seems to be a stronger inference than is warranted, given that there’s no pre-/post-test showing that lay people consider the low-calorie diet snacks significantly healthier than the “healthy” snacks (and/or lower in calories than the “healthy” snack), so I’m not convinced that participants necessarily thought of ONLY the “healthy” snacks as healthy but thought of the low calorie snacks as “not healthy but at least low-calorie.”

Another issue to consider re: this juxtaposition is that being healthy and being low-calorie is, in

practice, NOT mutually exclusive, even though the study appears to set it up that way. In many settings the two will be closely aligned (e.g., the healthiest options at a fast food restaurant are probably also the lowest calorie ones [e.g., non-fried preparation options, fruit/salad as sides, water for a drink]), and lay people tend to equate healthy and low-calorie.

4. Boundary/moderator conditions: What do the authors hypothesize might turn off the effect? Underlying this question is what the authors believe to be the key ingredient in nature exposure that drives the effect. Although it may not be necessary to examine the process in detail, at least some hypothesizing and a test of a related boundary condition would be theoretically and practically useful—it's hard to believe that any and all nature exposure produces these effects. Is it about seeing nature thrive and grow that inspires people to want to tend to and grow their body? In that case, dead/dying/devastated nature should not produce the effect. Is it about nature evoking awe (as Bellew and Omoto 2018 suggest), concomitantly leading to greater reverence of one's body, which facilitates healthier choices? If so, more mundane (vs. awesome) vistas should produce differently sized effects. These are just some examples to illustrate what sort of possible moderation I'm talking about.

Minor:

5. Results presentation in Study 2: Since nature increases healthy choices (rather than urban decreasing healthy choices), the results in Study 2 could be described differently. Currently, they say the % of healthy "drops" from nature to urban/control, which implies that urban/control is doing the work. Saying that the % of healthy choices is "boosted" in the natural condition may better reflect the authors' point.

6. Power analysis prior to Study 5: Good idea to do a power analysis to determine sample size. However, it'd be great if the text explained what the power analysis was based on (i.e., the effect sizes from the preceding 4 studies reported in THIS manuscript? Or rather the statistics from the inconclusive study from the other paper?).

7. Details on foods/snacks: Including info on what the exact snacks/foods were within the main text would be helpful, rather than having to go to the appendix. I also couldn't find any info on the snacks used in Study 4 (the one that investigated preference for healthy vs. low-calorie snacks).

Best of luck with this work!

Reviewer #3 (Remarks to the Author):

This high-quality manuscript includes five experiments: one field walking experiment and four online well-standardized studies. Together, the studies bring a coherent message and bring new insights in the effects of nature on dietary choices. The studies are rather simple in design (not strong as separate study), but each bring cumulative and complementary evidence: e.g., study 2 includes a neutral condition, study 3 explicitly tests the importance of subjective food healthiness, study 4 distinguishes in healthy versus low-calorie, study 5 replicated an earlier inconclusive study, and the studies test in different nationalities.

A limitation is the between-subject design (vs a within-subject design) and the lack of more in-depth data besides the main outcome. Related to that, I noticed the authors have sometimes a limited amount of descriptive data that should be used for group comparisons (see comment lists).

The online studies have a very large sample size, allowing detection at 95% power, while the first field experiment had no power calculation and is only done in 39 participants.

The literature gives a comprehensive overview of existing studies and the relevance of the study.

The studies and results are clearly described (except some details: see comment list). The discussion is to the point but rather short and does not consider weaknesses (see comment list).

Specific comments:

- The abstract could specify better from how many countries (multiple -> three) and the contexts ('such as' seems to say that there are more contexts than the 2 listed, although these are the only 2). Since all studies are between-subject, that seems also relevant to be mentioned in the abstract.

- Introduction:

- o A similar specification would be helpful at the start of the discussion: three food choice settings, three nationalities,... In that listing 'hold across a variety of foods, beverages, contexts, and nationalities' it seems like beverages were also tested separately, while that was not the case.

- o In the study goals, the authors mention 'in terms of food quality and/or quantity', while the discussion is not mentioning the quantity.

- o Hypothesis 5 (last sentence of intro) should be reformulated since it is now stating the hypothesis as a result (it demonstrates that...) rather than a hypothesis.

- o The authors have a nice, tabulated overview of existing studies, but probably <https://doi.org/10.1371/journal.pone.0176028> is still a relevant one missing.

- Results:

- o Please also to Table 2 the sample size and whether it is an online study or not.

- o Figure 2 would benefit of mentioning the sample sizes per group.

- o The last sentence of the results section interprets the results of study 5. The authors say that the power is the reason why they now find something that was not detected before. It could also be because of the dietary method since now the authors forced participants to one food choice only. Also a pity that the authors did not attempt to reproduce also that delay discounting.

- Discussion:

- o The discussion mentions within the mechanisms of action also that 'taste goals' are less driving the food choices. Based on the data these taste goals were not explicitly measured, so it would be best to make such statements with more caution.

- o In that same paragraph, the authors mention affect stress and self-perception as mechanism, but the concept of inhibition or delay discounting is missing (although it was mentioned in the introduction)

- o The discussion highlights the strengths but not the weaknesses. Despite the strong data, some weaknesses exist.

- Methods:

- o Study 1: Since a real-life experiment is less standardized than an online experiment and since it is a between-subject design, we would need to get some more information about how comparable the city and park group was: weather conditions, same moment of the day (afternoon snacking is more frequent), stress, any food vending along the path,...

- o Study 1: the authors mention that both environments were equally familiar to the participants: what is meant exactly, has this been verified?

- o Study 2: a limitation is that in the menu choice, the three choices (main course, side, and beverage) were not tested separately. This can be mentioned as exploratory analysis to give more insight on something that has not been tested before in literature

- o Study 3-4-5: From the preregistration and online material it is clear that some descriptive data has been collected: dieting behaviour, age, residential environment. Please check whether these characteristics differ between your two groups.

- o Appendix A mentions that naturalness was reported in the pre-test but no results were given. Please mention also those results in the appendix.

RESPONSES TO REVIEWER 1

Reviewer #1 (Remarks to the Author):

This series of studies aims to examine whether being immersed in and exposed to images of nature, as opposed to urban environments, encourages adults to make healthier food choices. The researchers conclude that this is true across all of their studies. They also conclude that the mechanism for this is an increase in motivation to make healthy eating choices after exposure to nature (as opposed to reducing calories, and as opposed to urban settings increasing motivation to make unhealthy eating choices).

Results could have implications for our understanding of the health benefits of nature and possibly developing effective interventions for healthier eating down the line. The researchers include some careful controls in these studies as well. However, I would want to re-review the paper to determine its suitability for publication with a few questions answered first:

Thank you very much for your constructive and detailed comments. Below, we summarize how we incorporated them into the new version of the manuscript.

Intro:

I would take out the beginning quote and also the bit about fresh air in the Intro since that is not what is being measured here. Both walking conditions provided fresh air, and then viewing conditions provided no fresh air, so there was no difference in fresh air exposure between environmental conditions. Plus, sometimes indoor air can also be healthier than outdoor air (ie wildfires of late), so this premise may be best to leave out altogether.

We agree and have removed the beginning quote and the bit about fresh air in the introduction.

Instead of “regular” eating, maybe use “typical” eating? (Intro)

Thank you for your suggestion. We have replaced “regular” eating with “typical” eating.

What is meant by “incentive-compatible food choice”?

We have clarified (p.10) that, by incentive-compatible food choice, we mean a “procedure with consequential decisions—where participants are incentivized to select foods based on their own true preferences, as there is a chance they will win the food they select”.

Methods

Did the authors make any measurements of stimuli/environment restorativeness? Ie, did they disentangle environmental category (natural vs urban) from perceived restorativeness? (ie Berman et al., 2008; Kotabe et al., 2017, Mahamane et al., 2020). If not, there should be some discussion on how it may not be nature itself that motivates healthy eating decisions, but perhaps some other dimension(s) not yet measured or disentangled here (that may correlate with environmental category).

You make a good point here, as research demonstrates that natural environments tend to be rated as more restorative on dimensions such as fascination, coherence, being-away, compatibility, and scope^{1,2}. In the updated version of the paper, we clarify that nature exposure is the intervention, not the psychological mechanism that explains its effects, and have incorporated “perceived restorativeness” in our suggestions for future research (p.12): “Future research is necessary to examine the role that affect, stress, perceived restorativeness, and self-perception may play in explaining why nature exposure increases the motivation for healthy eating”.

*Similarly, since Nature, Urban, and Control were often represented by only one picture—how do the researchers know that it was the environmental *category* influencing decisions, and not the subject of the individual particular photo the researchers chose to represent the category (ie a lake representing Nature in Studies 2 and 3 vs the word “Stop” on concrete representing the category Urban?)*

You are right to point out that studies 2 and 3 operationalize the urban environment through a photo that depicts an urban environment with a road that has “stop” written on the ground in the opposite direction. To circumvent issues relating to the choice of any specific photo, different photos were used for studies 4 and 5 (and for the additional study that we now report in Appendix E). We also address this concern through our field experiment in study 1, where participants do not see any photos but rather take a walk in an urban or natural environment.

To strengthen our claim that our effects are not tied to photos, we conducted an additional study (p.17) in “Appendix D: Implicit association task (IAT)”. This study shows an implicit association between nature-related (vs. urban-related) words (without images) and healthiness-related words.

Also in Study 2/3, how long did the participants look at the photos for?

In studies 2 and 3, participants were not confined to a specific length of time to look at the photographic stimuli. We know that the nature and urban photos were visible for at least the amount of time it took participants to write the one-sentence summary description of the scene. Additionally, participants re-encountered the photos in the survey and were asked to pay careful attention to the photographic scenes depicted, as these studies were initially framed as memory and visual perception pre-tests.

Rain was mentioned for one participant in Study 1—did participants generally experience the same (overall balance of) weather/temperature/time of day across conditions in Study 1? This would be important to know.

We agree that this is an important question. Study 1 occurred during the spring season in Paris, France, from late March to early/mid-May. Sessions were not scheduled on days when rain was predicted on the weather forecast, and only one session was canceled due to rain. To address your question, we obtained historical information on the weather,

temperature, wind, humidity, and barometric pressure from <https://www.timeanddate.com/weather> and matched them to the day and time each participant did the study. There was no statistically significant difference between the nature and urban groups on the temperature ($t(37)=-.87, p=.39$), degree of cloud/sun coverage ($t(37)=.05, p=.96$), wind ($t(37)=-.81, p=.42$), humidity ($t(37)=.39, p=.70$), nor barometric pressure ($t(37)=.41, p=.68$). The degree of cloud/sun coverage was coded as such: fog=1, overcast=2, mostly cloudy=3, broken clouds=4, passing clouds=5, scattered clouds=6, partly sunny=7, sunny=8.

Results

In Study 3, how did perceived healthiness align with actual healthiness?

This is a great question. To address it, we compared participants' own healthiness ratings in study 3 with an objective measure of the nutritional quality of the food, as measured via the FSA nutrient profiling model³. This score considers energy density, saturated fats, sugars, salt, proteins, dietary fibers, and the percentage of fruit and vegetables per 100 g or 100 ml. It ranges from -15 to +40, with a lower FSA score indicating higher nutritional quality. We used the FSA score because it is the measure of choice in epidemiological studies on the effects of food nutritional quality on health⁴. We obtained the FSA score of the products used in study 3 from <https://world.openfoodfacts.org>.

We found a significant correlation between the measures of objective and subjective healthiness ($r=-.84, p<.001$). As a result, all the effects of exposure to nature replicated when using this objective measure of nutritional quality. The code for these additional analyses is available in the repository. We would also be happy to add the results of these analyses to another appendix in the paper if you think it would be helpful.

Discussion

Other possible mechanisms like priming, attention, emotion, prosociality, intention to not harm environment, self-controlled decision-making, etc need more discussion. There is a lot of research on how nature can affect these, and so it may not be easy to discount interconnected mechanisms.

We agree entirely. Given that the paper focuses on establishing the effects of nature exposure rather than its psychological mechanisms, we did not elaborate on all the potential mediators. Instead, we wrote (p. 12), "Nature exposure's influence on healthy eating is likely multiply-determined. Therefore, we recommend that future research test multiple potential mechanisms simultaneously to compare their importance and the conditions under which they operate".

I am not really sure what this series of studies can tell us about the importance of outdoor recess in schools. Some schools would have urban and some natural environments for recess, and effects of indoor environments were not tested here as a comparison.

We have removed our statement on considerations for recess.

In sum, this is an important, fascinating study on a topic that contains constructs often difficult to operationalize. The authors rightly point out limitations and shortcomings. Overcoming these limitations would certainly increase impact of the study, and yet, I understand this is outside of the scope of this particular and still impressive series of studies. Thus, in the end, I do believe this could be appropriate for publication with a sufficient future revision.

We greatly appreciate all of the time, care, and attention that you invested in our work. Thank you for sharing your helpful feedback with us.

RESPONSES TO REVIEWER 2

Reviewer #2 (Remarks to the Author):

In the manuscript “Healthy by Nature: How Experiencing Nature Drives Healthy Food Choices” 5 studies show that exposure to nature boost propensity to choose healthy foods. Even though other research has started to investigate the nature—healthy eating connection, the paper clearly describes how its methodological design elements (e.g., experimentation, urban and control comparison groups) add to existing studies on the general link between nature and healthy eating. On the one hand, the field study with real nature experiences and food choice is impressive, and yet on the other hand it’s also valuable to know that the effect generalizes even to “fake” nature exposure (e.g., pictures on a computer screen). This suggests solid potential practical usefulness. Aside from these positives, there are a couple of conceptual and empirical questions the manuscript would benefit from clarifying.

Thank you very much for your helpful comments.

1. Analysis presentation and decisions could be clearer: It’s cumbersome to follow the analyses, perhaps because the designs are mixed and inherently more complex, with varying interdependencies (e.g., intervention is between-subjects but main/side/beverage choices are nested within-subject) but also because some statements describing what exact analyses were performed are vague (“accounting for the fact that each participant provided 8 observation”—how?). It may also be beneficial to explain the underlying rationales for choosing certain analysis strategies (e.g., what’s the advantage of treating each food choice as 0 vs. 1 in a design where people must choose exactly 1 from each category, compared to adding up the number of healthy items and running an ANOVA). To be clear, I’m NOT suggesting the authors chose inappropriate analysis strategies—I’m merely saying they could help readers better understand the value of the selected analyses.

Thank you for bringing this to our attention. To demonstrate the robustness of the results, we take three analytical approaches for study 1, where we initially provide descriptive statistics and analyses for the total quantity of healthy and unhealthy foods consumed. Next, we used a linear regression with the number of servings as the dependent variable—and used a random-effect model to allow for correlated errors at the participant level to account for the fact that each participant provided 8 observations (one per snack). This allows us to understand how exposure to nature and food healthiness interact to influence the food quantity consumed. Finally, we incorporated an analysis with the 0 vs. 1 design that you referenced—as it allows a direct comparison of these results to those of other studies that did not incorporate food quantity consumption (and focus exclusively on food quality, or 0 vs. 1 choice outcomes).

We also chose not to count the number of healthy choices in studies 2, 3, and 5 because that approach would result in a count between 0 and 3 as the dependent variable, which would not be an accurate depiction of the experiment’s choice architecture and the decision-making that participants are engaging in—ultimately violating the ANOVA assumption of a

dependent variable measured at the continuous level. Instead, we used a statistical model that matched with the decision required from the participants, where they were asked to choose 1 main course out of 4 options, 1 side dish out of 4 options, and then 1 drink out of 4 options. We therefore used a conditional logistic regression, which measures whether an option's attribute (i.e., healthy vs. unhealthy) can predict whether it is selected from the set of options. To allow for a comparison with the results of study 1 (and following our pre-registration), we also provide the results of a simpler analysis, a binary logistic regression for each of the 12 choices.

To better explain the value of our analyses, we have added the following sentence in the description of study 1 (p. 4): *“To test the effects of the intervention on the consumption of healthy and unhealthy food, we estimated a random effect model in SPSS using the MIXED procedure, which allows for correlated errors at the participant level to account for the fact that each participant provided 8 observations (one per food).”* We have also added the number of observations in each experimental cell.

2. *Role of healthiness motives: Study 3 claims to show that “exposure to nature drives individuals to make healthier food choices by giving rise to a motive to eat more healthfully” (p. 10). Apologies if I’m overlooking something, motives (or goals or whatever construct best encapsulates people’s reasons for doing something) were not measured anywhere. If by “importance” the authors mean the higher regression coefficients, then the statement is a bit tautological, as betas definitionally encode importance (albeit statistical importance to the regression, not necessarily psychological importance to people’s motivations).*

We agree, and have removed the sentence.

3. *Distinction between healthy vs. low-calorie is unclear: Two aspects of the question if nature inspires a preference for healthier vs. lower-calorie (examined in Study 4) remained unclear: (a) What exactly do the authors mean by healthy when they juxtapose it with low-calorie? I understand and agree that healthy and low-calorie are not equal and definitely distinguishable (e.g., from a nutritional quality standpoint, a snack of raw almonds will almost certainly be deemed healthier despite being more calorific than a snack of sugar-free chocolate). But what do the authors mean: more nutrient-dense or (as some phrasing on p. 10 suggests) more natural, as opposed to less nutrient-dense or more processed? Or something else?*

This is a very good point. “Low-calorie” and “healthy” are not the same but are not mutually exclusive either. We take a broad view of food healthiness that encompasses both nutritional quality and the food's level of processing (or naturalness). We have therefore clarified in the first paragraph of the paper (p. 2) that by “healthier” food, we mean “more nutritious and less processed” food.

Speaking to the distinction between calories and overall healthiness, study 4 showed that exposure to nature (vs. urban) images increases the choice of a “natural, healthy snack” but not of “a diet, light snack”. We cannot further disentangle nutrition quality and level of processing because the healthy food options used in the five studies were, as in real life, both more nutritious and less processed than the indulgent food options. However,

following the suggestion of another reviewer, we replicated the results of study 3 using an objective measure of nutritional quality rather than the participant's own perception of the food's healthiness. This measure, the FSA nutrient profiling score, does not consider the food's processing level. Given the choice of food stimuli, the results appear robust to different measures of a food's healthiness.

(b) Why do the authors feel this distinction is particularly important? Does it reveal something about the process? In the GD, one phrase seems to imply that this finding shows that the effect is "really" about healthy eating, not "just" about calorie reduction (p. 11, second to last paragraph). That seems to be a stronger inference than is warranted, given that there's no pre-/post-test showing that lay people consider the low-calorie diet snacks significantly unhealthier than the "healthy" snacks (and/or lower in calories than the "healthy" snack), so I'm not convinced that participants necessarily thought of ONLY the "healthy" snacks as healthy but thought of the low calorie snacks as "not healthy but at least low-calorie."

This is another excellent point. In study 4, we assumed but had not explicitly tested whether people consider the low-calorie snack significantly less healthy than the "natural, healthy snack". Therefore, we conducted a post-test, where we asked 100 online adult participants (U.S.-based, English-speaking) via Prolific to rate the 3 descriptions (i.e., "a natural, healthy snack", "a diet, light snack", and "a tasty, indulgent snack") on healthiness (7-point scale from "not at all healthy" to "extremely healthy"). We obtained N=99 valid participants due to 1 attention check failure. The "natural, healthy snack" is rated most healthy ($M=6.14$, $SD=1.00$), followed by the "diet, light snack" ($M=4.63$, $SD=1.36$), and finally, the "tasty, indulgent snack" ($M=2.32$, $SD=1.25$). Paired t -tests show that the "natural, healthy snack" is perceived as significantly healthier than the "diet, light snack" ($t(98)=10.38$, $p<.001$, $d = 1.45$) and significantly healthier than the "tasty, indulgent snack" ($t(98)=25.31$, $p<.001$, $d = 1.50$). The "diet, light snack" is also perceived as significantly healthier than the "tasty, indulgent snack" ($t(98)=14.38$, $p<.001$, $d = 1.59$). We added these results in Appendix C.

Another issue to consider re: this juxtaposition is that being healthy and being low-calorie is, in practice, NOT mutually exclusive, even though the study appears to set it up that way. In many settings the two will be closely aligned (e.g., the healthiest options at a fast food restaurant are probably also the lowest calorie ones [e.g., non-fried preparation options, fruit/salad as sides, water for a drink]), and lay people tend to equate healthy and low-calorie.

We agree, so now we explain that we do not view the different interpretations of healthy food as mutually exclusive (p. 10): "Multiple and non-mutually exclusive interpretations of food healthiness coexist. For example, low-calorie "diet" food and unprocessed nutritious foods are similarly marketed as "healthy", even though prior research has documented large differences in food choices depending on whether people's motivation is health per se or the desire to lose weight".

4. Boundary/moderator conditions: What do the authors hypothesize might turn off the effect? Underlying this question is what the authors believe to be the key ingredient in nature exposure that drives the effect. Although it may not be necessary to examine the process in detail, at least

some hypothesizing and a test of a related boundary condition would be theoretically and practically useful—it's hard to believe that any and all nature exposure produces these effects. Is it about seeing nature thrive and grow that inspires people to want to tend to and grow their body? In that case, dead/dying/devastated nature should not produce the effect. Is it about nature evoking awe (as Bellew and Omoto 2018 suggest), concomitantly leading to greater reverence of one's body, which facilitates healthier choices? If so, more mundane (vs. awesome) vistas should produce differently sized effects. These are just some examples to illustrate what sort of possible moderation I'm talking about.

We agree that studying the boundary conditions for the effect is interesting. We had already conducted one such study, which we did not include in the previous version of the paper because of concerns about the length of the paper. We have included this study as Appendix E, but would gladly add it as study 6 in the paper if the editorial team recommends it.

This study establishes that the effects of nature exposure disappear when the photos of nature and urban scenes are taken in the winter with snow, which is consistent with your prediction that “dead” nature should not produce these effects. Because we agree that there could be other boundary conditions, we also added this sentence (p. 12): “Additionally, we recommend that future work explore the boundary conditions associated with nature’s effects on healthy eating. Appendix E reports the results of a study showing a similar proportion of healthy food choices when the photos of nature or urban scenes were taken in the winter and included snow, suggesting that greenery may play a role, although the characteristics of the landscape itself may also matter⁵. It would be interesting to explore whether certain elements of vitality or awe could be altered to enhance or suppress nature’s influence on food choice.”

Minor:

5. Results presentation in Study 2: Since nature increases healthy choices (rather than urban decreasing healthy choices), the results in Study 2 could be described differently. Currently, they say the % of healthy “drops” from nature to urban/control, which implies that urban/control is doing the work. Saying that the % of healthy choices is “boosted” in the natural condition may better reflect the authors’ point.

Thank you for this helpful suggestion. We implemented your recommendation, and we now report the following (pg. 8), “In the urban condition, 62% of the selected foods were healthy, and in the control condition, 65% of the selected foods were healthy (see Figure 2); that proportion increased to 71% in the nature condition ($\chi^2=13.76, p<.01$).”

6. Power analysis prior to Study 5: Good idea to do a power analysis to determine sample size. However, it'd be great if the text explained what the power analysis was based on (i.e., the effect sizes from the preceding 4 studies reported in THIS manuscript? Or rather the statistics from the inconclusive study from the other paper?).

We have clarified this (p. 15), “As with the preceding studies, the number of participants was determined to achieve 95% power with a two-sided $\alpha = 0.05$ based on the results of study 2.”

7. Details on foods/snacks: Including info on what the exact snacks/foods were within the main text would be helpful, rather than having to go to the appendix. I also couldn't find any info on the snacks used in Study 4 (the one that investigated preference for healthy vs. low-calorie snacks).

We apologize for the lack of clarity. Study 4 did not use visual stimuli; instead, we used text to describe the three potential snack options. Accordingly, we provide the following in our methods section for Study 4 (p. 15), “Study 4 did not include photos of specific foods but asked participants to choose between three textual descriptions : “a natural, healthy snack”, “a tasty, indulgent snack”, or “a diet, light snack”.”

Best of luck with this work!

We greatly appreciate all of the time, care, and attention that you invested in our work. Thank you for sharing your helpful feedback with us.

RESPONSES TO REVIEWER 3

Reviewer #3 (Remarks to the Author):

This high-quality manuscript includes five experiments: one field walking experiment and four online well-standardized studies. Together, the studies bring a coherent message and bring new insights in the effects of nature on dietary choices. The studies are rather simple in design (not strong as separate study), but each bring cumulative and complementary evidence: e.g., study 2 includes a neutral condition, study 3 explicitly tests the importance of subjective food healthiness, study 4 distinguishes in healthy versus low-calorie, study 5 replicated an earlier inconclusive study, and the studies test in different nationalities.

Thank you very much for your helpful comments.

A limitation is the between-subject design (vs a within-subject design) and the lack of more in-depth data besides the main outcome. Related to that, I noticed the authors have sometimes a limited amount of descriptive data that should be used for group comparisons (see comment lists). The online studies have a very large sample size, allowing detection at 95% power, while the first field experiment had no power calculation and is only done in 39 participants. The literature gives a comprehensive overview of existing studies and the relevance of the study. The studies and results are clearly described (except some details: see comment list). The discussion is to the point but rather short and does not consider weaknesses (see comment list).

Specific comments:

- The abstract could specify better from how many countries (multiple -> three) and the contexts ('such as' seems to say that there are more contexts than the 2 listed, although these are the only 2). Since all studies are between-subject, that seems also relevant to be mentioned in the abstract.

We appreciate these helpful points about the abstract. To address your first point, we changed “from multiple countries” to “from three countries”. To your second point, we have replaced “such as” with “comprising of”. In response to your final point, we added “between subjects” to the sentence “Five between-subjects experiments”.

- Introduction:

o A similar specification would be helpful at the start of the discussion: three food choice settings, three nationalities, ... In that listing 'hold across a variety of foods, beverages, contexts, and nationalities' it seems like beverages were also tested separately, while that was not the case.

We agree that beverages were not tested separately from food, but we did test multiple types of healthy and unhealthy beverages. To make this clearer, we have grouped beverages together with foods, where the statement reads as, “... for a variety of foods and beverages, across samples from three countries...”. In this, we feel that we're expressing that various foods and beverages have been used as outcome variable stimuli, as opposed to claiming separate testing.

o In the study goals, the authors mention ‘in terms of food quality and/or quantity’, while the discussion is not mentioning the quantity.

Thank you for raising this valid point. To address your comment, we included the following statement in our discussion section (p. 12), “We also recommend that future work challenge and explore what it means to eat “healthy” in this context, as elements like food quantity add an additional layer of nuance and understanding to healthy food choice”.

o Hypothesis 5 (last sentence of intro) should be reformulated since it is now stating the hypothesis as a result (it demonstrates that...) rather than a hypothesis.

Good point. We rephrased the sentence (p. 4) that you are referring to by framing the study as an exploration of “prior inconclusive results that may have been driven by a lack of power and/or the reliance on indirect measures of food desirability rather than more direct measures of food choices”.

o The authors have a nice, tabulated overview of existing studies, but probably <https://doi.org/10.1371/journal.pone.0176028> is still a relevant one missing.

Thank you for this very helpful reference. We have included it in our tabulated overview of existing studies (p. 3).

- Results:

o Please also to Table 2 the sample size and whether it is an online study or not.

We have added both elements to Table 2. “Summary of studies” (p. 5).

o Figure 2 would benefit of mentioning the sample sizes per group.

We added the number of participants in each experimental condition in the main text (methods section). Due to space constraints, we were not able to add sample sizes in the graphs for Figure 2 without significantly distorting the images.

o The last sentence of the results section interprets the results of study 5. The authors say that the power is the reason why they now find something that was not detected before. It could also be because of the dietary method since now the authors forced participants to one food choice only. Also a pity that the authors did not attempt to reproduce also that delay discounting.

We agree that the dietary method used in the inconclusive study could also explain why it failed to find an effect. We had written (p. 10), “We hypothesize that the null results in prior studies were caused by a lack of power or sensitivity in the measures that led to type II errors.” To further underscore this point, we now repeat it in the general discussion (p. 11): “previous inconclusive results were driven by a lack of statistical power or sensitivity in their measure of food preferences”. While we also agree that measuring delay discounting would have been interesting too, it is beyond the scope of our research.

- Discussion:

o The discussion mentions within the mechanisms of action also that 'taste goals' are less driving the food choices. Based on the data these taste goals were not explicitly measured, so it would be best to make such statements with more caution.

We agree that we did not measure taste goals (only healthfulness ratings) and have rephrased the sentence in the general discussion as follows (p. 11), “we found that exposure to nature increases the importance of health in driving food choices while decreasing preferences for reduced-calorie or indulgent foods”.

o In that same paragraph, the authors mention affect stress and self-perception as mechanism, but the concept of inhibition or delay discounting is missing (although it was mentioned in the introduction).

Good point. We have added delay discounting to our recommendations (p. 12) for future research.

o The discussion highlights the strengths but not the weaknesses. Despite the strong data, some weaknesses exist.

You are right to point this out. As a result, we have added the following to our discussion section (p. 12): “One of the weaknesses in our research is that we did not determine how long the benefits derived from exposure to nature endure. Another weakness is that we only studied the food choices made for a single consumption occasion, such as a snack or lunch. To address this issue, it would be important to conduct longitudinal research looking at the effects of nature exposure on changes in diet over time.”

- Methods:

o Study 1: Since a real-life experiment is less standardized than an online experiment and since it is a between-subject design, we would need to get some more information about how comparable the city and park group was: weather conditions, same moment of the day (afternoon snacking is more frequent), stress, any food vending along the path,...

We agree that this is an important point. Study 1 took place during the spring season in Paris, France, from late March to early/mid-May. Sessions were not scheduled on days when rain was predicted on the weather forecast, and only one session was canceled due to rain.

To address your question, we obtained historical information on the weather, temperature, wind speed, humidity, and barometric pressure from <https://www.timeanddate.com/weather> and matched them to the day and time each participant did the study. The degree of cloud/sun coverage was coded as such: fog=1, overcast=2, mostly cloudy=3, broken clouds=4, passing clouds=5, scattered clouds=6, partly sunny=7, sunny=8. There was no statistically significant difference between the nature and urban groups on the temperature ($t(37)=-.87, p=.39$), degree of cloud/sun coverage

$(t(37)=-.05, p=.96)$, wind $(t(37)=-.82, p=.42)$, humidity $(t(37)=.95, p=.35)$, nor barometric pressure $(t(37)=.41, p=.68)$.

o Study 1: the authors mention that both environments were equally familiar to the participants: what is meant exactly, has this been verified?

It is true that we did not measure participants' familiarity with the locations and have hence replaced "... and were equally familiar to the participants", with, "... and had the same starting point". Nevertheless, we think the urban and nature walks were familiar because they originated in a popular university and cultural center and included either a popular public park or a prominent street directly across from the starting point.

o Study 2: a limitation is that in the menu choice, the three choices (main course, side, and beverage) were not tested separately. This can be mentioned as exploratory analysis to give more insight on something that has not been tested before in literature.

We pre-registered that we would combine the three choices (main course, side dishes, and beverages) because it allowed us to gather more observations and increase the statistical power of the analyses, and because there was no theoretical rationale for expecting different results across the three choices. In addition, we think that studying the effects separately for each choice would be problematic because it would ignore the compensation effects⁶ that occur when people create a meal (e.g., people choosing a healthier dessert because they had chosen an indulgent main course). For these reasons, we did not test the effects separately, but the data are available on ResearchBox for other researchers who wish to explore these issues.

o Study 3-4-5: From the preregistration and online material it is clear that some descriptive data has been collected: dieting behaviour, age, residential environment. Please check whether these characteristics differ between your two groups.

Thank you for this suggestion. We examined carefully whether these variables moderated the effects of nature exposure but did not find any robust pattern across studies. However, demographic and other individual variables are available on ResearchBox for others to explore.

o Appendix A mentions that naturalness was reported in the pre-test but no results were given. Please mention also those results in the appendix.

We apologize for this omission. We made sure to include these results (p. 16) in Appendix A: "The naturalness ratings mirrored those of the healthiness ratings. The healthy foods group was evaluated as higher in overall naturalness ($M=6.06, SD=0.66$) when compared to the unhealthy foods group ($M=2.45, SD=0.94, t=73.4, p<.001$)."

We greatly appreciate all of the time, care, and attention that you invested in our work. Thank you for sharing your helpful feedback with us.

REFERENCES

1. Berto R. Exposure to restorative environments helps restore attentional capacity. *Journal of environmental psychology* 2005, **25**(3): 249-259.
2. Korpela K, Hartig T. Restorative qualities of favorite places. *Journal of environmental psychology* 1996, **16**(3): 221-233.
3. Julia C, Kesse-Guyot E, Touvier M, Mejean C, Fezeu L, Hercberg S. Application of the British Food Standards Agency nutrient profiling system in a French food composition database. *British Journal of Nutrition* 2014, **112**.
4. Donnenfeld M, Julia C, Kesse-Guyot E, Mejean C, Ducrot P, Peneau S, *et al.* Prospective association between cancer risk and an individual dietary index based on the British Food Standards Agency Nutrient Profiling System. *British Journal of Nutrition* 2015, **114**(10): 1702-1710.
5. Michels N, Debra G, Mattheeuws L, Hooyberg A. Indoor nature integration for stress recovery and healthy eating: A picture experiment with plants versus green color. *Environmental Research* 2022: 113643.
6. Abou Jaoudé L, Denis I, Teyssier S, Beugnot N, Davidenko O, Darcel N. Nutritional labeling modifies meal composition strategies in a computer-based food selection task. *Food Quality and Preference* 2022, **100**: 104618.

4th Jan 24

Dear Dr Langlois,

Your manuscript titled "Healthy by Nature: How Experiencing Nature Drives Healthy Food Choices" has now been seen by our reviewers, whose comments appear below. In light of their advice I am delighted to say that we are happy, in principle, to publish a suitably revised version in Communications Psychology under the open access CC BY license (Creative Commons Attribution v4.0 International License).

We therefore invite you to revise your paper one last time to address the remaining concerns of our reviewers and a list of editorial requests. At the same time we ask that you edit your manuscript to comply with our format requirements and to maximise the accessibility and therefore the impact of your work.

EDITORIAL REQUESTS:

SUBMISSION INFORMATION:

OPEN ACCESS:

Communications Psychology is a fully open access journal. Articles are made freely accessible on publication under a CC BY license (Creative Commons Attribution 4.0 International License). This license allows maximum dissemination and re-use of open access materials and is preferred by many research funding bodies.

For further information about article processing charges, open access funding, and advice and support from Nature Research, please visit <https://www.nature.com/commspsychol/article-processing-charges>

At acceptance, you will be provided with instructions for completing this CC BY license on behalf of all authors. This grants us the necessary permissions to publish your paper. Additionally, you will be asked to declare that all required third party permissions have been obtained, and to provide billing information in order to pay the article-processing charge (APC).

* TRANSPARENT PEER REVIEW: Communications Psychology uses a transparent peer review system.

On author request, confidential information and data can be removed from the published reviewer reports and rebuttal letters prior to publication. If you are concerned about the release of confidential data, please let us know specifically what information you would like to have removed. Please note that we cannot incorporate redactions for any other reasons.

* CODE AVAILABILITY: All Communications Psychology manuscripts must include a section titled "Code Availability" at the end of the methods section. We require that the custom analysis code supporting your conclusions is made available in a publicly accessible repository at this stage; please choose a repository that generates a digital object identifier (DOI) for the code; the link to the repository and the DOI must be included in the Code Availability statement. Publication as Supplementary Information will not suffice.

* DATA AVAILABILITY:

[link redacted]

Best regards,

Marike, on behalf of Hannah Hao

Marike Schiffer, PhD
Chief Editor
Communications Psychology

REVIEWERS' COMMENTS:

Reviewer #1 (Remarks to the Author):

I am satisfied with the authors' response, and recommend publication. This is a revision with much improvement!

Reviewer #2 (Remarks to the Author):

The authors delivered a responsive and thoughtful revision. The analyses are now described much more clearly, as are the distinction between healthy and low-calorie choices (and I especially appreciate the post-test data) and the basis for the power analysis. Moderators are discussed, and I suppose that is sufficient given that the paper is chiefly about the overall effect of nature exposure on healthy choices as opposed to the underlying mechanism. These changes address my previous concerns about the manuscript, which studies an interesting question and offers a distinct contribution to the literature on food preference and consumption behavior.

Reviewer #3 (Remarks to the Author):

The authors have responded adequately to all questions, and adapted the manuscript accordingly. Some limitations of course remain, but the article improved sufficiently. Nice that the authors included new data on nature versus city during winter. However, since this winter city picture includes a lot of water/ocean, it seems not a perfect control situation. This should be addressed as limitation.

RESPONSES TO REVIEWERS

Reviewer #1 (Remarks to the Author):

I am satisfied with the authors' response, and recommend publication. This is a revision with much improvement!

Thank you very much for your helpful feedback through the manuscript review process. We truly appreciate the time and attention that you dedicated towards helping us improve our manuscript.

Reviewer #2 (Remarks to the Author):

The authors delivered a responsive and thoughtful revision. The analyses are now described much more clearly, as are the distinction between healthy and low-calorie choices (and I especially appreciate the post-test data) and the basis for the power analysis. Moderators are discussed, and I suppose that is sufficient given that the paper is chiefly about the overall effect of nature exposure on healthy choices as opposed to the underlying mechanism. These changes address my previous concerns about the manuscript, which studies an interesting question and offers a distinct contribution to the literature on food preference and consumption behavior.

We greatly appreciate all of the time and attention that you dedicated towards helping us improve our paper. Thank you very much for all of your helpful feedback throughout the manuscript review process.

Reviewer #3 (Remarks to the Author):

The authors have responded adequately to all questions, and adapted the manuscript accordingly. Some limitations of course remain, but the article improved sufficiently. Nice that the authors included new data on nature versus city during winter. However, since this winter city picture includes a lot of water/ocean, it seems not a perfect control situation. This should be addressed as limitation.

We are pleased to know that we have adequately addressed all of your concerns. We appreciate the time and attention that you committed towards helping us improve our manuscript.

Regarding the additional seasonality-focused data, you are right to point out that one of the winter-urban scenes contains water; it is worth noting that participants viewed two sets of photos per experimental condition, where we ensured that the other winter-urban scene did not contain any water.

Thanks again for all of your helpful comments and insights. We greatly appreciate it.